# Investigating the Diversity of *Wolbachia* across the Spiny Ants (*Polyrhachis*)

Jenna L. Webb [1] , Leland C. Graber [1], Manuela O. Ramalho [1,2,*] and Corrie S. Moreau [1,3]

1 Department of Entomology, Cornell University, Ithaca, NY 14580, USA
2 Department of Biology, West Chester University, West Chester, PA 19393, USA
3 Department of Ecology and Evolutionary Biology, Cornell University, Ithaca, NY 14850, USA
* Correspondence: mramalho@wcupa.edu; Tel.: +1-312-874-8751

**Abstract:** Among insects, *Wolbachia* is an exceedingly common bacterial endosymbiont with a range of consequences of infection. Despite the frequency of *Wolbachia* infection, very little is known about this bacteria's diversity and role within hosts, especially within ant hosts. In this study, we analyze the occurrence and diversity of *Wolbachia* across the spiny ants (*Polyrhachis*), a large and geographically diverse genus. *Polyrhachis* samples from throughout the host genus' phylogenetic and biogeographical range were first screened for single infections of *Wolbachia* using the *wsp* gene and Sanger sequencing. The multilocus sequence typing (MLST) scheme was then used on these singly infected samples to identify the *Wolbachia* strains. A *Wolbachia* phylogeny was inferred from the *Polyrhachis* samples analyzed in this study as well as other Formicidae MLST profiles from the MLST online database. We hypothesized that three key host factors were impacting *Wolbachia* diversity within the *Polyrhachis* genus: biogeography, phylogeny, and species level. The results suggest that the phylogeny and biogeography of *Polyrhachis* hosts have no impact on *Wolbachia* diversity; however, species level may have some limited influence. Additionally, *Wolbachia* strains appear to group according to being either Old World or New World strains. Among the taxa able to form complete MLST allelic profiles, all twenty are seemingly new strains.

**Keywords:** multilocus sequence typing (MLST); Formicidae; host–microbe associations





## 1. Introduction

Ants (Formicidae) are a highly diverse family of insects with a global distribution. One of the many factors contributing to the overwhelming ecological success of ants is their many associations with symbiotic microbes. Ants associate with microbial eukaryotes, fungi, viruses, and bacteria; further, many of these associations are understood to have contributed to the diversity of diets, occupied niches, and life history in various ant groups [1]. For example, the ant genus *Cephalotes* is able to survive on a nutrient-poor herbivorous diet due to the microbial symbionts present in its gut [2]. In addition, the functions of these symbionts are of particular interest to researchers, especially in the case of maternally transmitted bacterial symbionts already known to alter host reproduction, development, nutrition, and defense in many arthropods [3]. The *Wolbachia* bacterial genus is a well-known example of such a symbiont. It is estimated that *Wolbachia* infects up to 75% of all insect species [4] and is an incredibly common, heritable maternally transmitted bacterial symbiont of ants [3,5]. Some of the most notable consequences of *Wolbachia* infection in insects are alterations to the host's reproductive abilities—these include parthenogenesis, male killing, male feminization, and cytoplasmic incompatibility [6]. Within ants (Formicidae) specifically, *Wolbachia* has also been found to accelerate the colony life cycle [7] and enhance the host's nutrient uptake [8]. Due to the variety of *Wolbachia's* impacts on its ant hosts, more studies are needed to elucidate the diversity of *Wolbachia* across Formicidae to understand the consequences of its associations with ants [9,10].

While it is now known that *Wolbachia* is a widespread symbiont of insects, it was first discovered as a rickettsial symbiont of the mosquito *Culex pipiens* in the 1920s [11]. All *Wolbachia* strains are divided into supergroups via phylogenetic analysis using one or multiple marker genes (e.g., 16S rDNA, *wsp, ftsZ*). Currently, there are twenty-one *Wolbachia* supergroups, ranging from A to U [11–13]. Further, these *Wolbachia* supergroups also appear to have set associations to specific host taxa. For instance, it has been found that the strains in Formicidae hosts are from mostly supergroups A and F with the majority being from supergroup A [9], though there has been a single instance where a supergroup B strain was found associated to an ant host from Mexico, *Pheidole sciophila* [14].

Previously, the standard procedure for sequence typing *Wolbachia* strains was based upon sequencing the *Wolbachia* surface protein gene, *wsp* [15]. After it was determined that *wsp* experiences extensive recombination via swapping of conserved amino acid motifs within hyper-variable regions [16], the Multilocus Sequence Typing (MLST) approach was proposed [6]. MLST was introduced alongside an online database of bacterial and host information (https://pubmlst.org/organisms/wolbachia-spp, accessed on 27 February 2023), and *Wolbachia* sequence types are based upon the allele determination of five different housekeeping genes (*coxA, fbpA, ftsZ, gatB,* and *hcpA*) rather than *wsp* [4]. MLST has become the standard method of sequence typing *Wolbachia* since it provides a more robust approach to assessing *Wolbachia* diversity across a variety of host taxa due to the reliance on five loci rather than the *wsp* locus alone. The MLST scheme has been used by researchers studying this bacterial genus within a wide range of hosts including filarial nematodes and ticks [17], butterflies [18] and ants [1,19–21].

*Polyrhachis* Smith, 1857 is a large ant genus (over 700 species) that inhabits Africa, Asia, Australia, and Oceania [22,23]. They are commonly called "spiny ants" due to the spinescence of most species, which can vary in shape, length, and numbers; this spinescence is hypothesized to be a defense characteristic against vertebrate and invertebrate predators [24,25]. They exhibit a large variety of nesting techniques including the use of larval silk to weave their nests (a trait limited to few ant genera), nesting inside hollow bamboo, and attaching nests to stones [26,27]. In addition, *Polyrhachis* belongs to the Camponotini tribe, which is well known for their symbiotic relationships with bacteria—in particular *Blochmannia* [28]—and an association with *Wolbachia* has been previously found in Camponotini as well [20,29,30]. *Polyrhachis's* broad biogeographical range spanning across Africa, Asia, Australia, and Oceania [26] makes it a useful host for studying the impacts of host biogeography on *Wolbachia* diversity.

In past studies, specific ant species have been studied for their associations to *Wolbachia* [19,23], and the evolutionary association of *Wolbachia* was evaluated across the entire Formicidae family [9]. Additionally, *Wolbachia* has been studied in other social insects such as bees, termites, and wasps [10]. In one species-specific study, the diversity of *Wolbachia* was analyzed across the geographically diverse giant turtle ant species (*Cephalotes atratus*), and results suggested that *Wolbachia* diversity is affected by geography [19]. In a broader study that analyzed *Wolbachia* across Formicidae, the evolutionary origins of *Wolbachia* infection in ants were illuminated and the biogeographical origin of the symbiosis was inferred to be in Asia [9]. Our intention with this work, investigating *Wolbachia* infection across the *Polyrhachis* genus, is to further explore the notion that *Wolbachia* diversity can be impacted by geography, as well as the evolutionary association between host and microbe via phylogenetic and species level analyses.

In the following study, our primary objective was to analyze and observe patterns of *Wolbachia* infection in *Polyrhachis*. We hypothesized that three factors related to the *Polyrhachis* host will impact the observed diversity of *Wolbachia*: phylogeny, species level, and biogeography. If these factors do impact *Wolbachia* diversity, we would anticipate seeing significant correlations in increases (or decreases) of *Polyrhachis* host diversity with that of its *Wolbachia* symbionts. If, for example, the *Polyrhachis* host phylogeny impacts the observed diversity of *Wolbachia*, we will see phylogenetic signal and potential evolutionary co-diversification between *Polyrhachis* and *Wolbachia.* If species level within *Polyrhachis*

impacts the observed diversity of *Wolbachia*, we may observe different kinds of *Wolbachia* infecting different *Polyrhachis* species in statistically significant ways. If biogeography of the *Polyrhachis* host impacts the observed diversity of its *Wolbachia*, we may see a variance in *Wolbachia* infection that is correlated with the different locations where each *Polyrhachis* sample was collected.

## 2. Materials and Methods

Samples from 102 *Polyrhachis* species (237 *Polyrhachis* samples) were screened for their associated *Wolbachia* strains. These samples were collected from 29 countries (Table 1 shows samples positive for *Wolbachia*; all samples are listed in Supplementary Material File S1). The DNA extractions was performed on whole ant specimens following the DNeasy Blood and Tissue (Qiagen) protocol. The DNA was stored at $-20$ °C. The sampled *Polyrhachis* species were taken to be representative of the entire host genus and spanned across the entire *Polyrhachis* biogeographical range. To screen for *Wolbachia* and determine which samples contained single infections, sequencing of *Wolbachia*'s *wsp* gene was performed. The *wsp* gene was PCR amplified using Taq DNA Polymerase, primers wsp81f and wsp69r (at 1 μM each), and 1 μL of DNA [17,31] for 36 cycles with an annealing temperature of 59 °C [4]. The thermocycler program was set to the following: the cycle began with denaturation at 94 °C for 30 s, annealing for 45 s, 72 °C for 1.5 min, an elongation step at 70 °C for 10 min, and a hold at 4 °C. Annealing temperatures varied by gene: *coxA* was annealed at 55 °C, *fbpA* at 59 °C, *hcpA* at 53 °C, and both *ftsZ* and *gatB* at 54 °C. The PCR products were first evaluated using gel electrophoresis [32] wherein the presence of a band indicated the infection of at least one *Wolbachia* strain for that sample. *Wolbachia*-positive PCR products were purified using ExoSap (Cleveland, OH, USA) with the manufacturer-recommended thermocycler settings. BigDye Terminator (Applied Biosystems, Waltham, MA, USA) was used to prepare the samples for Sanger sequencing, which was carried out by the Cornell Institute of Biotechnology (Ithaca, NY, USA). The resulting sequence electropherograms were evaluated in Geneious Prime 2022.1 (https://www.geneious.com, accessed on 15 August 2022) to determine whether samples were infected with single or multiple strains of *Wolbachia*.

**Table 1.** Sample ID, host species, and country of origin for all samples positive for *Wolbachia*. A complete list of samples screened is available in Supplementary Material File S1.

| Sample ID | Species | Country | Sample ID | Species | Country |
|---|---|---|---|---|---|
| DG06 | *(Polyrhachis (Myrmatopa)* sp. | Phillipines | RA0766 | *Polyrhachis flavibasis* | Australia |
| ISR_06 | *Polyrhachis (Myrma)* sp. | Thailand | SUL02 | *Polyrhachis (Myrma)* sp. 1 | Indonesia |
| GM 894 | *Polyrhachis (Myrmhopla)* sp. 2 | Malaysia | SKY20 | *Polyrhachis* sp. | Singapore |
| GM3990 | *Polyrhachis (Myrmhopla)* sp. 4 | Malaysia | SL_28_2 | *Polyrhachis illaudata* | Malaysia |
| GM3589b | *Polyrhachis (Myrmothrinax)* sp. | Malaysia | SKY24 | *Polyrhachis* sp. | Singapore |
| AS4132a | *Polyrhachis (Polyrhachis)* sp. | Cambodia | LEA04 | *Polyrachis schistaceae* | Mozambique |
| CSM0776 | *Polyrhachis abbreviata* | Australia | MS1177 | *Polyrhachis shixingensis* | China |
| DG10 | *Polyrhachis armata* | Philippines | RA0784 | *Polyrhachis* sp. | Solomon Islands |
| DG14 | *Polyrhachis armata* | Phillipines | RA1157 | *Polyrhachis illaudata* | Laos |
| CSM0761 | *Polyrhachis australis* | Australia | MJ9286 | *Polyrhachis* sp. | Papua New Guinea |
| DG26 | *Polyrhachis bicolor* | Philippines | RA1163 | *Polyrhachis illaudata* | Laos |
| BB012 | *Polyrhachis bihamata* | China | MJ 8277 | *Polyrhachis* sp. | Papua New Guinea |
| CSM1806a | *Polyrhachis bihamata* | Malaysia | PH09 | *Polyrhachis afrc_cd03* | Democratic Republic of the Congo |
| CSM1806b | *Polyrhachis bihamata* | Malaysia | PH11 | *Polyrhachis laboriosa* | Democratic Republic of the Congo |
| DG08 | *Polyrhachis bihamata* | Phillipines | RA0769 | *Polyrhachis "chario5"* | Australia |
| CSM1846 | *Polyrhachis boltoni* | Malaysia | PH14 | *Polyrhachis gagates* | South Africa |
| EMS2584 | *Polyrhachis campbelli* | Solomon Islands | RA736a | *Polyrhachis dives-group* sp. | Thailand |
| DG04 | *Polyrhachis carbonaria* | Phillipines | RA0765 | *Polyrhachis ammon* | Australia |
| CSM1854 | *Polyrhachis cephalotes* | Malaysia | PH15 | *Polyrhachis afr_cd01* | Democratic Republic of the Congo |
| EMS2617 | *Polyrhachis* cf. *bismarckensis* | Solomon Islands | RA736c | *Polyrhachis* cf. *laevissima* | Thailand |
| CSM1841 | *Polyrhachis danum* | Malaysia | PH12 | *Polyrhachis revoili* | Democratic Republic of the Congo |
| BB28 | *Polyrhachis hippomanes* | China | MJ 9243 | *Polyrhachis* sp. *near bicolor* | Papua New Guinea |
| JRNG01 | *Polyrhachis hookeri* | Australia | TAS 02 | *Polyrhachis hexacantha* | Australia |

**Table 1.** *Cont.*

| Sample ID | Species | Country | Sample ID | Species | Country |
|---|---|---|---|---|---|
| DG03 | *Polyrhachis illaudata* | Phillipines | RA0755 | *Polyrhachis "BATH3"* | Australia |
| GM3551 | *Polyrhachis illaudata* | Malaysia | SKY21 | *Polyrhachis nigropilosa* | Singapore |
| DG25 | *Polyrhachis inermis* | Philippines | RA1162 | *Polyrhachis illaudata* | Laos |
| EMS2637 | *Polyrhachis kaipi* | Solomon Islands | RO 122 | *Polyrhachis* sp. | Tanzania |
| ISR_03 | *Polyrhachis lacteipennis* | Israel | SOH 02 | *Polyrhachis beccari* | Singapore |
| CSM1868 | *Polyrhachis lepida* | Malaysia | PH21 | *Polyrhachis schistacea* | Mozambique |
| DG16 | *Polyrhachis near lilianae* | Philippines | RA1158 | *Polyrhachis mucronata-group* sp. | Laos |
| BB48 | *Polyrhachis proxima* | China | MJ 9280 | *Polyrhachis mucronata-group* sp. | |
| CSM0655 | *Polyrhachis rufifemur* | Australia | MJ8280 | *Polyrhachis* sp. | Papua New Guinea |
| CSM0740 | *Polyrhachis rufifemur* | Australia | MJ9242 | *Polyrhachis sexspi-sa group* | Papua New Guinea |
| DG11 | *Polyrhachis saevissima* | Phillipines | RA1154 | *Polyrhachis mucronata-group* sp. | Laos |
| DG17 | *Polyrhachis saevissima* | Phillipines | RA1160 | *Polyrhachis illaudata?* | Laos |
| KATE02 | *Polyrhachis schistacea* | South Africa | TAS04 | *Polyrhachis semipolita* | Australia |
| AS4132b | *Polyrhachis* sp. | Cambodia | LEA05 | *Polyrachis schistaceae* | Mozambique |
| BB026 | *Polyrhachis* sp. | China | MJ 8282 | *Polyrhachis sexspi-sa group* | Papua New Guinea |
| CSM1860 | *Polyrhachis* sp. | Malaysia | PH16 | *Polyrhachis latharis* | Democratic Republic of the Congo |
| CSM2632 | *Polyrhachis* sp. | Uganda | PH22 | *Polyrhachis schistacea* | Tanzania |
| CSM2738 | *Polyrhachis* sp. | Uganda | PSW5403 | *Polyrhachis andromache* | Australia |
| CSM2745 | *Polyrhachis* sp. | Uganda | PSW6454 | *Polyrhachis obesior* | Malaysia |
| CSM2831 | *Polyrhachis* sp. | Australia | RA0735 | *Polyrhachis abdominalis* | Singapore |
| FH1085 | *Polyrhachis* sp. | Uganda | RO538 | *Polyrhachis* sp. | Tanzania |
| FH1101 | *Polyrhachis* sp. | Uganda | SKY05 | *Polyrhachis frustorferi* | Indonesia |
| FH205 | *Polyrhachis* sp. | Kenya | SKY11 | *Polyrhachis lamellidens* | Japan |
| FH987 | *Polyrhachis* sp. | Uganda | SKY17 | *Polyrhachis hector* | Indonesia |
| JCM120P | *Polyrhachis* sp. | Palau | TAS 01 | *Polyrhachis hexacantha* | Australia |
| JRNG02 | *Polyrhachis* sp. | Australia | TAS03 | *Polyrhachis phryne* | Australia |
| LD01 | *Polyrhachis* sp. | Ghana | LEA03 | *Polyrachis schistaceae* | Mozambique |
| AS4121 | *Polyrhachis* sp. *near furcata* | Cambodia | MJ 8263 | *Polyrhachis* sp. | Papua New Guinea |
| AS4148a | *Polyrhachis* sp. *near furcata* | Cambodia | MJ 8291 | *Polyrhachis* sp. | Papua New Guinea |
| BB_075 | *Polyrhachis* sp. *near sixspi-sa* | China | MJ9275 | *Polyrhachis* sp. | Papua New Guinea |
| CSM0746 | *Polyrhachis thais* | Australia | SL32 | *Polyrhachis furcata* | Malaysia |
| IND05 | *Polyrhachis thrinax* | India | MJ 9287 | *Polyrhachis* sp. | Papua New Guinea |
| CAB01 | *Polyrhachis ypsilon* | Malaysia | | | |

　　　　Only singly infected *Polyrhachis* samples (n = 34) were subjected to the Multilocus Sequence Typing (MLST) process wherein the five MLST genes (*coxA*, *fbpA*, *ftsZ*, *gatB*, and *hcpA*) were amplified and sequenced according to the same procedure as was done for the *wsp* gene. Since Sanger sequencing of multiple strains at once creates indecipherable electropherograms (due to the sequences for each strain overlaying each other), multi-infected *Polyrhachis* samples were excluded from the MLST process. Sequence alignments for each locus were created in Geneious Prime 2022.1 (https://www.geneious.com, accessed on 15 August 2022) then checked against reference sequences in the MLST online database (https://pubmlst.org/organisms/wolbachia-spp, accessed on 15 August 2022) to determine closest matching allele types. The closest matching sequence type (ST) for each *Wolbachia* strain able to produce clear electropherograms for all five loci (n = 20) was determined based upon these five alleles.

　　　　The five MLST genes were concatenated (2098 bp total length, order: *coxA*, *fbpA*, *ftsZ*, *gatB*, *hcpA*) for each of the 20 remaining samples, then added to a pool of 70 MLST database sequences from other Formicidae-associated *Wolbachia* strains [33]. A *Wolbachia* phylogeny was inferred with these 90 MLST sequences via the IQ-Tree web server 1.6.12 [34] to infer a phylogenetic tree by maximum likelihood and generate bootstrap values. The best fit model of substitution for each locus was determined by the ModelFinder [35] and partition model [36] features available through IQ-Tree web server. Partitions and their best-fit models are shown in Table 2. *Wolbachia* strains for ST124 and ST557 (both supergroup F) from the host species *Ocymyrmex picardi* and *Paratrechina*, respectively, formed the outgroup

of the phylogeny. The haplotype network for each MLST gene was constructed with Network 4.5.1.0 [37] using the median joining parameter.

**Table 2.** Partitioning and best-fit models for each partition as determined by the IQ-Tree web server ModelFinder. The third and fourth columns detail the length of each partition and its position within the concatenated sequence (with a total length of 2098 bp).

| Partition | Gene(s) | Position in Concatenation (bp) | Length of Gene(s) in Partition (bp) | Model |
|---|---|---|---|---|
| 1 | *coxA* | 1–403 | 403 | HKY + F+G4 |
| 2 | *fbpA* | 404–840 | 437 | HKY + F+G4 |
| 3 | *hcpA, ftsZ* | 1651–2098, 841–1277 | 448, 437 | TIM3 + F+I+G4 |
| 4 | *gatB* | 1278–1650 | 373 | TIM + F+I+G4 |

Two mantel tests were performed using the R package vegan [38]. The first of these tested the correlation between phylogenetic distance between *Wolbachia* strains and geographical distances between latitudes and longitudes of collection sites. The second test examined correlations between phylogenetic distances between *Polyrhachis* host species (Blanchard and Moreau, in press) and phylogenetic distances of *Wolbachia* strains. Both the *Wolbachia* and *Polyrhachis* phylogenies were pruned down to seven tips, representing ant host or *Wolbachia* from seven different *Polyrhachis* species: *P. bihamata, P. cephalotes, P. carbonaria, P. thrinax, P. shixigensis, P. illaudata,* and *P. hexacantha.*

## 3. Results

Of the initial 237 *Polyrhachis* samples screened for *Wolbachia* using the *wsp* gene, 112 (47%) tested positive (Table 3). Positive samples represent 69 of the 102 tested *Polyrhachis* species. To test the hypothesis of host phylogeny influence on *Wolbachia* diversity, in the subsequent analyses we kept only the *Wolbachia*-positive samples of host species present in the *Polyrhachis* phylogeny generated by Mezger and Moreau [24]; this reduced the sample size to 73 samples. There were 43 different *Polyrhachis* species across the 73 samples. After analyzing the electropherograms to evaluate if the positive samples were single or multiple infections of *Wolbachia*, 34 of the 73 samples (47%) were determined to be single infections. Single and multiple infections of *Wolbachia* in *Polyrhachis* occurred in the same six biogeographical regions (Figure 1A); the singly infected samples were collected from 15 different countries (Figure 1B). Of the 34 singly infected samples, there were 21 different *Polyrhachis* species represented (Figure 1C).

**Table 3.** The allele, ST, and *Polyrhachis* host information for the 34 *Polyrhachis* samples. Allele numbers are included for each MLST gene that was able to be sequenced; blue-shaded alleles are close matches i.e., new allele variants for those loci. A dash (-) in an MLST gene column indicates that no sequence was able to be produced and thus no allele determination was made, and a dash in the ST column indicates a sample that was unable to be assigned to a sequence type. An asterisk (*) indicates the closest matching ST. The strains from samples CSM2738, MJ9280, MJ9287, MS1177, and SUL02 all had multiple "close matching" STs according to the MLST database, indicated by two asterisks (**). The country of origin and host species are shown in the last two columns.

| Sample ID | MLST Allele Number | | | | | ST | Country | *Polyrhachis* Host Species |
|---|---|---|---|---|---|---|---|---|
| | *coxA* | *fbpA* | *ftsZ* | *gatB* | *hcpA* | | | |
| AS4121 | 2 | 51 | 45 | 20 | 47 | 61 * | Cambodia | *P. (Myrmhopla)* sp. |
| AS4132b | 2 | 51 | 45 | 20 | 47 | 61 * | Cambodia | *P. (Polyrhachis)* sp. |
| AS4148a | 2 | 51 | 45 | 20 | 47 | 61 * | Cambodia | *P. (Myrmhopla)* sp. |
| BB012 | 2 | 51 | 45 | 20 | 47 | 61 * | China | *P. bihamata* |
| CSM1854 | 2 | 51 | 45 | 20 | 47 | 61 * | Malaysia | *P. cephalotes* |
| CSM2738 | 33 | 61 | 47 | 34 | 195 | ** | Uganda | *P. (Myrma)* sp. |
| DG04 | 2 | 51 | 45 | 20 | 47 | 61 * | Philippines | *P. carbonaria* |

**Table 3.** *Cont.*

| Sample ID | MLST Allele Number | | | | | ST | Country | *Polyrhachis* |
| | *coxA* | *fbpA* | *ftsZ* | *gatB* | *hcpA* | | | Host Species |
|---|---|---|---|---|---|---|---|---|
| GM3589b | 2 | 356 | 258 | 22 | 343 | 52 * | Malaysia | *P. (Mymothrinax)* sp. |
| IND05 | 2 | 52 | 45 | 20 | 47 | 61 | India | *P. thrinax* |
| MJ9243 | 2 | 51 | 45 | 20 | 47 | 61 * | Papua New Guinea | *P. (Myrmhopla)* sp. |
| MJ9280 | 33 | 465 | 17 | 3 | 343 | ** | Papua New Guinea | *P. (Myrmhopla)* sp. |
| MJ9287 | 33 | 463 | 17 | 130 | 343 | ** | Papua New Guinea | *Polyrhachis* sp. |
| MS1177 | 33 | 463 | 17 | 3 | 343 | ** | China | *P. shixingensis* |
| RA1157 | 2 | 51 | 45 | 20 | 47 | 61 * | Laos | *P. illaudata* |
| RA1163 | 2 | 51 | 45 | 20 | 47 | 61 * | Laos | *P. illaudata* |
| RA736c | 2 | 51 | 45 | 20 | 47 | 61 * | Thailand | *P. cf. laevissima* |
| RO122 | 2 | 51 | 45 | 20 | 47 | 61 * | Tanzania | *P. (Myrma)* sp. |
| SKY24 | 32 | 48 | 6 | 57 | 50 | 51 * | Singapore | *Polyrhachis* sp. |
| SUL02 | 296 | 97 | 258 | 3 | 343 | ** | Indonesia | *P. (Myrma)* sp. |
| TAS02 | 33 | 277 | 17 | 3 | 343 | 481 * | Australia | *P. hexacantha* |
| CSM0655 | 32 | - | 6 | 57 | 50 | - | Australia | *P. rufifemur* |
| DG11 | 2 | - | 258 | 22 | 343 | - | Philippines | *P. saevissima* |
| EMS2584 | 218 | 6 | - | 158 | 141 | - | Solomon Islands | *P. campbelli* |
| EMS2617 | 33 | - | 17 | 3 | 343 | - | Solomon Islands | *P. bismarckensis* |
| FH1101 | 2 | - | 261 | 20 | 47 | - | Uganda | *P. (Myrma)* sp. |
| GM894 | 32 | - | 6 | 57 | 50 | - | Malaysia | *P. (Myrmhopla)* sp. |
| KATE02 | - | - | - | - | - | - | South Africa | *P. schistacea* |
| MJ8291 | - | - | - | - | - | - | Papua New Guinea | *Polyrhachis* sp. |
| MJ9286 | 109 | - | 261 | 191 | 83 | - | Papua New Guinea | *Polyrhachis* sp. |
| RA0755 | 32 | - | - | 57 | 50 | - | Australia | *Polyrhachis "BATH3"* |
| RA0784 | 33 | - | 17 | 3 | 343 | - | Solomon Islands | *P. (Myrmatopa)* sp. |
| RA1158 | 2 | - | 258 | 182 | 343 | - | Laos | *P. (Myrmhopla)* sp. |
| RA1160 | 2 | - | 45 | 20 | 47 | - | Laos | *P. illaudata* |
| TAS03 | 33 | - | 17 | 3 | 343 | - | Australia | *P. phryne* |

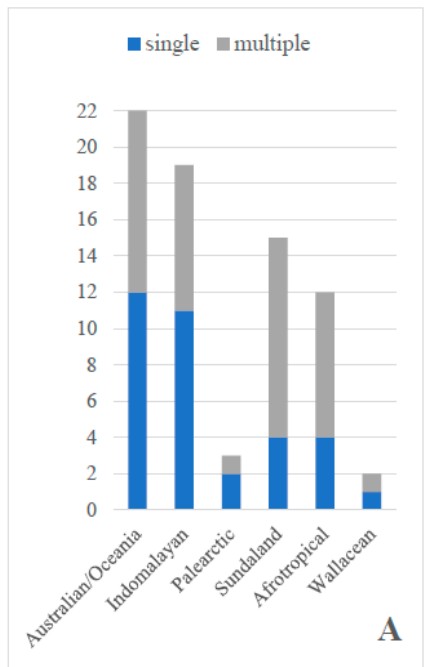

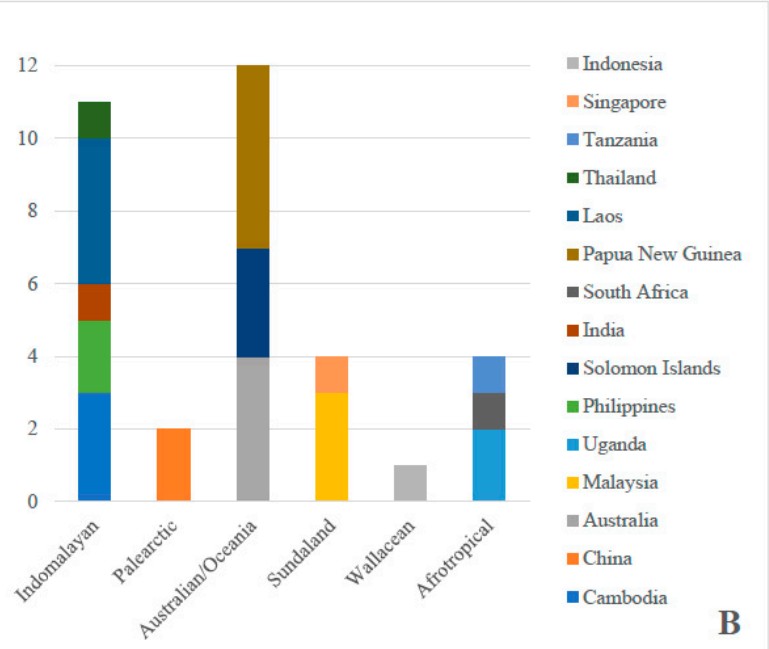

**Figure 1.** *Cont.*

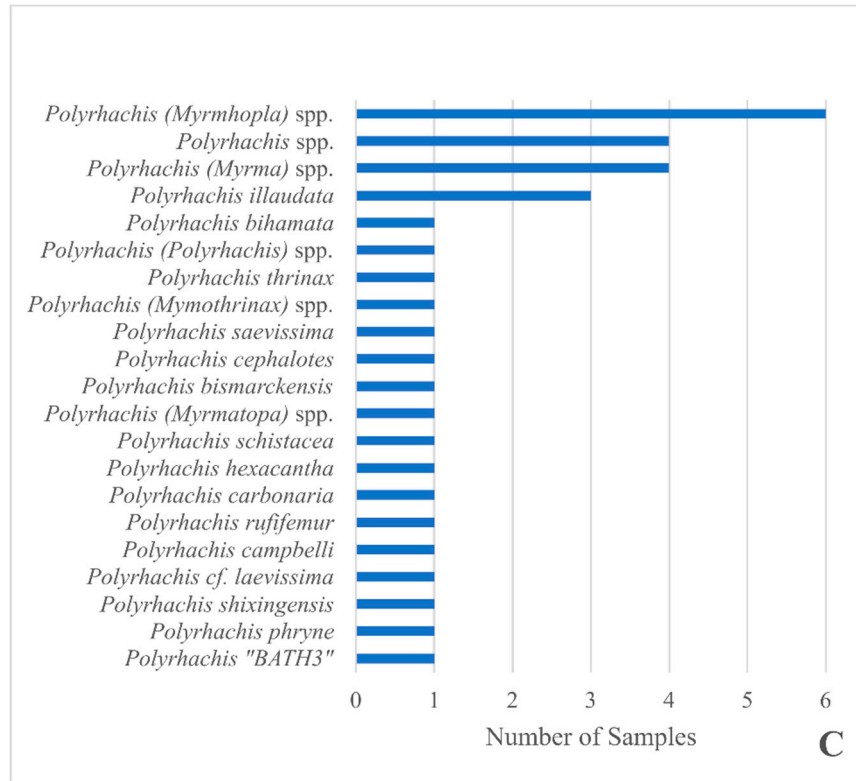

**Figure 1.** (**A**) Biogeographic distribution of single-strain *Wolbachia* infections and multi-strain *Wolbachia* infections within *Polyrhachis* hosts. The "Sundaland" and "Wallacean" groups are included here as separate categories to better distinguish their geography from the more northern parts of the Indomalayan realm. (**B**) Country distribution of the singly infected *Polyrhachis* samples. (**C**) *Polyrhachis* species distribution of the 34 singly infected samples. There were 21 different host species represented in this sample pool.

Twenty of the singly infected samples were able to produce viable sequences for all five MLST loci. Table 3 shows the allele and ST determinations, as well as host information, for those 20 samples. Further, Figure 2 illustrates nucleotide differences in the form of a haplotype network. Loci with no exact matches to sequences in the MLST database were considered to have new allele variants—every strain identified had at least two loci with new variants. The 14 singly infected samples that were unable to produce complete MLST alignments each had at least one locus with indeterminable Sanger results—two samples, KATE02 and MJ8291 (from *Polyrhachis shistacea* in South Africa and *Polyrhachis* sp. in Papua New Guinea, respectively) were unable to produce sequences for any of the five loci (Table 3).

The *coxA* and *gatB* loci were seemingly the most stable MLST loci for *Polyrhachis*-associated *Wolbachia* strains. Of the five loci, they had the most samples with exact matches to allele variants currently registered in the MLST database, with only five possible new allele variants found at both loci. The *ftsZ* and *hcpA* loci presented a greater number of new allele variants than either the *coxA* or *gatB* loci: seven and eight new allele variants were found at the *ftsZ* and *hcpA* loci, respectively. The *fbpA* locus presented the most genetic change of all five loci when compared to references in the *Wolbachia* MLST database—17 of the 20 samples with complete MLST profiles presented new allele variants, each of which appear to be unique. Additionally, 13 samples with incomplete MLST allelic profiles produced indeterminable sequences for the *fbpA* locus, and for 10 of these samples, *fbpA* was the only locus unable to be properly sequenced (Table 3). Ultimately, due to each strain having at least one new MLST allele variant, it appears that all 20 samples present new *Wolbachia* STs not yet seen in the MLST database.

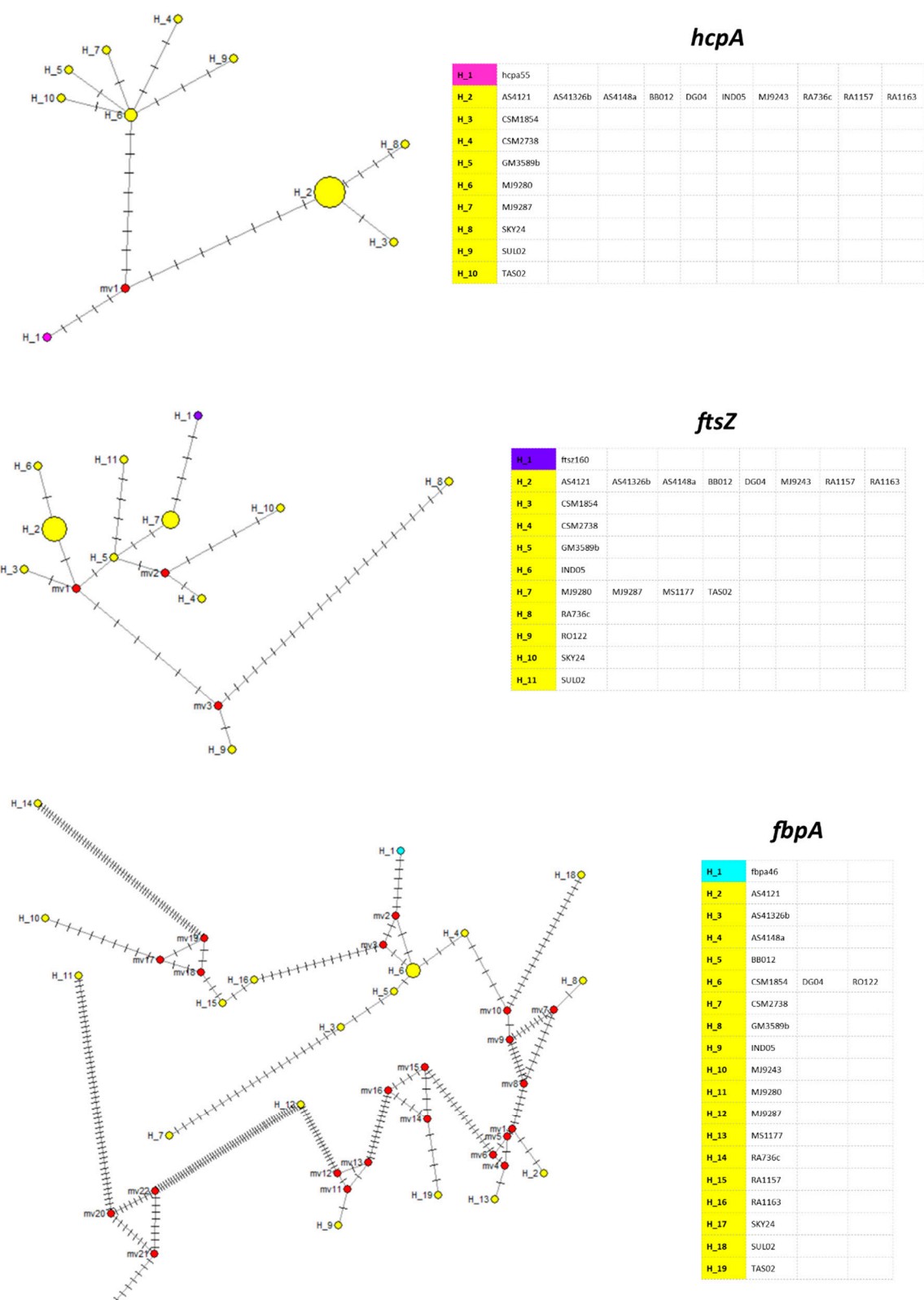

**Figure 2.** *Cont.*

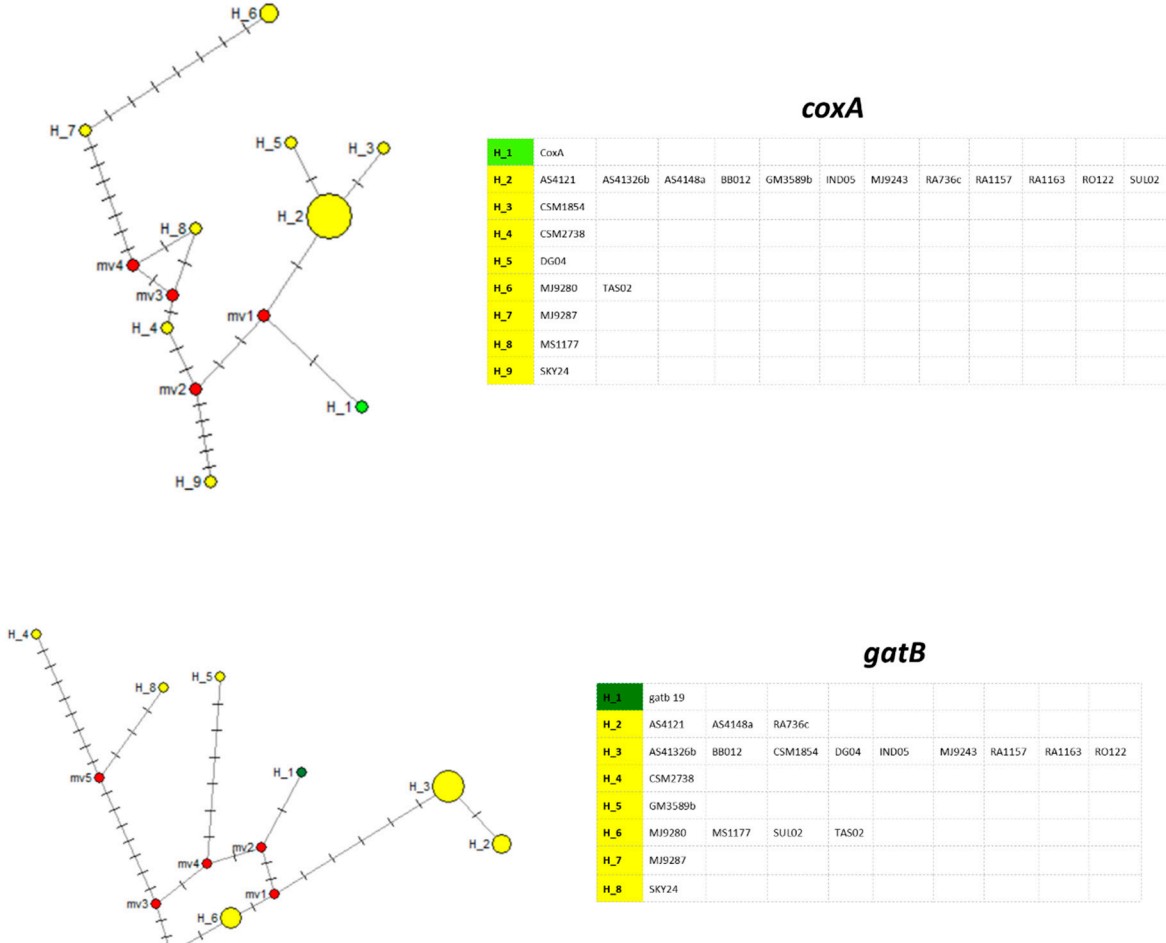

**Figure 2.** Haplotype network figures for all samples able to be assigned a sequence type. The haplotype size represents the frequency found, and black bars between haplotypes represent the numbers of nucleotide differences between haplotypes. Red dots (labelled "mv" and numbered) were added by the program as a hypothetical haplotype.

The phylogeny inferred with 90 *Wolbachia* MLST sequences (20 are the *Polyrhachis*-associated from this study, 70 are other Formicidae-associated strains from the MLST database) is shown in Figure 3. Bootstrap values $\leq$ 70% were hidden. No samples exhibited close relationships to any *Wolbachia* strains from the outgroup, supergroup F. Thus, all *Wolbachia* found in *Polyrhachis* belong the Supergroup A. The *Polyrhachis* strains from this study were organized into 13 genotypes, seven of which contain only one *Polyrhachis*-associated strain (either independently or with another Formicidae-associated strain). Only two of these genotypes contain samples from the same country of origin: *P.* (*Myrmhopla*) sp. and *P.* (*Polyrhachis*) sp. from Cambodia, and the two *P. illaudata* samples from Laos. Additionally, the samples from Laos are also the only grouping which contains *Wolbachia* strains from the same host species (*Polyrhachis illaudata*). In addition, all *Polyrhachis*-associated *Wolbachia* strains grouped with other *Polyrhachis*-associated strains, which suggests that there is a specificity of *Wolbachia* for *Polyrhachis* species. Six biogeographical ranges are represented in the phylogeny by the *Polyrhachis*-associated strains and all *Polyrhachis*-associated strains were grouped together with other Old World samples. Additionally, distinct clades formed to separate *Wolbachia* into Old World and New World groupings. The clades "a" and "c" contain several samples from the same biogeographical region–the Old World. Clade "b" are mixed, however contain two subclades: "b1" with samples from the Old World, and "b2" with samples from the New World (Figure 3).

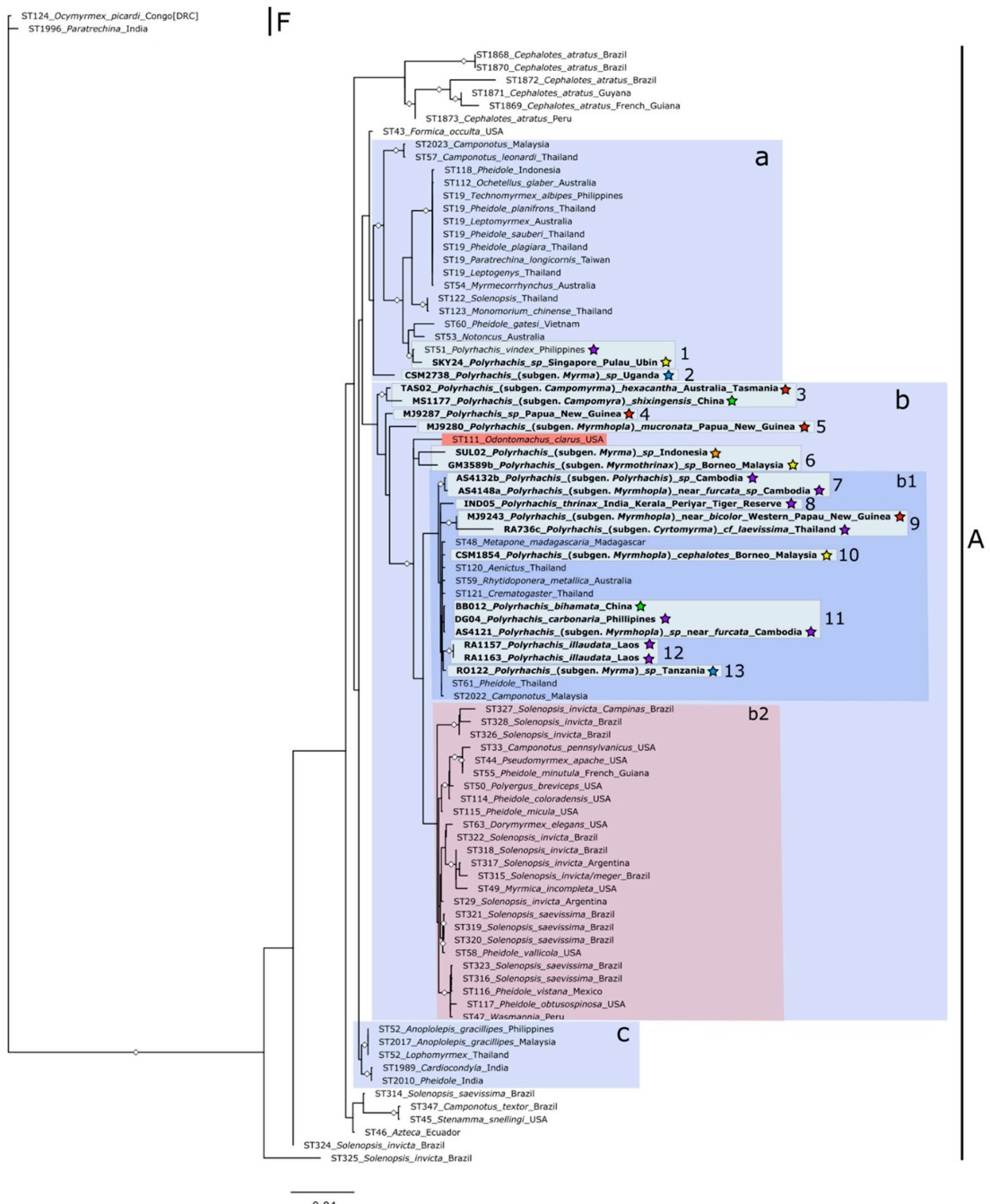

**Figure 3.** *Wolbachia* phylogeny. Samples are named here with the following convention: "Sequence-Type_Genus_species_Country". White diamonds indicate bootstrap values ≥ 70%. The bolded taxa are the 20 *Wolbachia* strains from this study; each of the 13 genotypes they formed are numbered and highlighted in light teal. All 20 taxa belong to the supergroup A clade. The supergroup F clade is the outgroup. Stars next to the taxa within each sample indicate biogeography: Afrotropical (n = 2, blue), Australian/Oceania (n = 4, red), Palearctic (n = 2, green), Indomalayan (n = 8, purple), Sundaland (n = 3, yellow), Wallacean (n = 1, orange). Only two of the 13 genotypes formed by the *Polyrhachis*-associated strains contain multiple samples from the same biogeographical region (seven and 12). Clades "a," and "c" are the Old World clades that formed within the supergroup A taxa. Within clade "b," "b1" represents Old World taxa while "b2" represents the New World taxa that seemingly evolved from Old World taxa. The red-highlighted taxon (ST111_*Odontomachus_clarus*_USA) is the only strain to not group according to being from an Old World or New World sample.

Results of the Mantel tests indicated no correlation between both *Wolbachia* phylogenetic distance and geographic distance (Mantel statistic r: −0.030; *p*-value: 0.531) and *Polyrhachis* phylogenetic distance and *Wolbachia* phylogenetic distance (Mantel statistic r: 0.117; *p*-value: 0.302).

## 4. Discussion

By using such a large and biogeographically diverse host genus like *Polyrhachis*, we were able to study whether host geography, phylogeny, and species level have any observed impact on *Wolbachia* diversity. Although 12 of the strains with complete allelic profiles best matched to ST61, they are all seemingly unique since their ST determination is based upon apparently new allele variants at multiple loci. For instance, RO122 (*Polyrhachis* [subgen. *Myrma*] sp. from Tanzania) was best matched to ST61 while having a possible variant at the *ftsZ* locus, but CSM1854 (*Polyrhachis cephalotes* from Malaysia) was also best matched to ST61 while having possible new variants at the *coxA*, *ftsZ*, and *hcpA* loci (Table 3). Indeed, these two samples were divided into their own clades in the *Wolbachia* phylogeny and there appears to be no tendencies for other strains with the same best matching STs to be grouped into clades. Therefore, our results suggests that each strain is a new ST (for a total of 20 new *Wolbachia* strains being found across the *Polyrhachis* genus), implying that across *Polyrhachis* there is an incredible diversity of *Wolbachia*.

The inferred *Wolbachia* phylogeny indicates that all strains identified in *Polyrhachis* aare from supergroup A, since there were no ant samples from this study that nested within the outgroup clade. Since the 20 strains included in the phylogeny span across the entire *Polyrhachis* geographic range, this phylogeny also suggests that the *Wolbachia* found within this host genus will likely belong to supergroup A, independent of the host's geographic range.

Some studies have seen that *Wolbachia* strains may group according to being Old World or New World [7,16], and it appears that the inferred phylogeny follows this trend as well. The blue boxes in Figure 3 represent Old World clades ("a," "b," and "c") that formed among the Supergroup A taxa—taxa not included in these boxes are strains from New World samplings. Clade "b" was further divided into clades "b1" and "b2"—"b1" being Old World taxa that seemingly evolved from Old World taxa, and "b2" being New World taxa that evolved from Old World taxa. Both taxa within the supergroup F outgroup are from Old World hosts. The only taxon that did not group according to the New World and Old-World clades is an ST111 strain from another study from an *Odontomachus clarus* host in the United States (highlighted red in Figure 3). This New World taxon grouped most closely into Old World clade "b1" and closest to *Polyrhachis* clade six. All other strains sourced from New World hosts formed exclusive New World clades. To understand why this *O. clarus* strain best fit into an Old World clade—and close to a *Polyrhachis* clade—rather than with other New World samples, more sampling of *Wolbachia* from that host genus would be necessary. Clades "a" and "b" appear to share a more recent common ancestor than they do with clade "c." Interestingly, all *Polyrhachis*-associated strains fell into the more closely related "a" and "b" clades; however, the majority were grouped into clade "b" with only clades one and two being part of "a." Ultimately, all but one taxon grouped according to being New or Old World, but it was not a perfect split-grouping since there were multiple clades of either type. Regardless, this still supports the trend seen in previous studies of *Wolbachia* [7,16] wherein strains will form clades according to Old or New World geography.

Among the 70 database MLST profiles used to infer the *Wolbachia* phylogeny, there was one strain also sourced from a *Polyrhachis* host (ST51_*Polyrhachis_vindex*_Philippines). This strain showed close relation to the strain from sample SKY24 (*Polyrhachis* sp. from Singapore), and together they form a distinct clade (clade one, Figure 3). However, since these samples are sourced from different hosts and different countries, this clade suggests that *Wolbachia* diversity is not significantly impacted by host species level or biogeography. Rather, this clade (as well as the other 12 clades) suggests that strains are likely to be more closely related if they are from the same host genus since no clades formed with strains

sourced from different host genera. Although they did not form a single, unified clade, the fact that all 13 clades contain exclusively *Polyrhachis*-associated strains suggests that the host's genus has some degree of influence on the associated *Wolbachia* diversity.

Of the 13 clades that the *Polyrhachis*-associated *Wolbachia* strains formed within the phylogeny, two had biogeographical consistency across the clade—clade 12 with two samples from Laos and clade seven with two samples from Cambodia—with both clades being from the Indomalayan biogeographical range (Figure 3). Interestingly, clade 12 contains the only two representative samples for the host species *Polyrhachis illaudata* (RA1163, RA1157), but based on their allelic profiles from Table 3 they are perhaps more likely to be closely related STs rather than the exact same STs. Although both strains were flagged as having possible new allele variants at the *fbpA* and *ftsZ* loci (and RA1163 with an additional variant at the *coxA* locus), they are not flagged for the same nucleotide modifications at either loci. Further verification of the genetic alterations that indicate these loci as having new allele variants would need to be conducted in order to distinguish these samples as different STs.

*Polyrhachis illaudata* was also the only host species able to have multiple samples with complete MLST allelic profiles sequenced. Both *P. illaudata*-associated strains formed a single clade (clade 12, Figure 3), suggesting that *Wolbachia* strains from the same host species will be more related than strains sampled from different host species. If all sampled *Polyrhachis* hosts receive expanded sampling across multiple colonies, it will be possible to determine whether this trend is true to other *Polyrhachis* hosts beyond *P. illaudata*. Thus, current results suggest that the species level of *Polyrhachis* hosts potentially impacts the observed *Wolbachia* diversity within this host genus.

The samples sourced from Cambodian *Polyrhachis* hosts present an interesting case. There was a third sample from Cambodia, AS4121 (*Polyrhachis* [subgen. *Myrmhopla*] sp.), not included in clade seven (Figure 3) with the other two Cambodian samples, AS4148a (*Polyrhachis* [subgen. *Myrmhopla*] sp.) and AS4132b (*Polyrhachis* [subgen. *Polyrhachis*] sp.)—this is seemingly because AS4148a and AS4132b both have the same new allele variant at the *gatB* locus whereas AS4121 has an already documented *gatB* allele variant (Table 3). Yet the host of AS4121 is more closely related to the host of AS4148a since they both belong to the subgenus *Myrmhopla*, while the host of AS4132 is subgenus *Polyrhachis* [24]. This instance suggests, then, that neither geography nor host phylogeny impacts the association of *Wolbachia* strains since more closely related hosts do not share similar *Wolbachia* strains and strains with hosts from the same country and geographical region do not appear in the same clade. Indeed, the Mantel test results support this since there was no correlation found between *Wolbachia* phylogenetic distance and geographical distance or between *Wolbachia* phylogenetic distance and *Polyrhachis* phylogenetic distance. However, this does not necessarily exclude the possibility that host species level could still be an impactor on *Wolbachia* diversity as seen in the *P. illaudata* clade that formed (clade 12, Figure 3).

Overall, the trends seen among the samples from Cambodia appear across the phylogeny—there is no consistent grouping of *Wolbachia* strains according to how related their host species are. For example, SUL02 and RO122 are both from subgenus *Myrma* of *Polyrhachis*, and MJ9280 and MJ9243 are both from subgenus *Myrmhopla*. Yet, in both cases, the two taxa are distantly related into two separate clades (SUL02 clade six, RO122 clade 13; MJ9280 clade 5, MJ9243 clade nine). From the perspective of host geography, there is rarely consistency for samples sourced from the same region to have more closely related strains. The three samples from Sundaland—GM3589b (*Polyrhachis* [subgen. *Myrmothrinax*] sp. from Malaysia), CSM1854 (*Polyrhachis cephalotes* from Malaysia), and SKY24 (*Polyrhachis* sp. from Singapore)—have perhaps the most distinct case of exhibiting that host geography may have no impact on the strain similarity of associated *Wolbachia*. Despite two of the three samples being from the same country, the three samples are split into three distant clades (6, 10, and one, respectively) in the phylogeny which, again, suggests that the geography of *Polyrhachis* hosts is not structuring *Wolbachia* diversity. Previous studies across genera in butterflies [18] and termites [39] similarly concluded that host geography did not impact which *Wolbachia* strains would be associated to the host. The study in termites also found that distantly related host species could have more

closely related *Wolbachia* strains [39] as seen in this study, thereby supporting the notion that the phylogeny of *Polyrhachis* hosts also has no strong impact on *Wolbachia* associations. In contrast, these results may contradict the results of Kelley et al. [19], which found that the association of *Wolbachia* to *Cephalotes atratus* was impacted by host biogeography. Yet, this may not be a true contradiction if it can be confirmed that across a single *Polyrhachis* species, host biogeography impacts *Wolbachia* diversity (which is seemingly seen in the *P. illaudata* clade [clade 12, Figure 3]) since the study by Kelley et al. [19] took place in a single host species.

For the third host factor (host species level), some results suggest that it has an impact on *Wolbachia* diversity. However, as discussed with the case of *P. illaudata*, expanded sampling of each *Polyrhachis* species is required to verify the observed trends. In the initial sample pooling, there were multiple instances where the same host species was sampled from several colonies. However, once removing samples containing multiple strains of *Wolbachia* the sample pool was reduced by over 50% and many of these multi-colony samplings were lost. These samples were removed because multiple strains in one sample cannot be parsed into individual strains.

We found that *Polyrhachis*-associated *Wolbachia* strains will form exclusive clades distinct from strains of other host genera. In other words, *Polyrhachis*-sourced strains of *Wolbachia* will only form clades with other *Polyrhachis*-associated strains. It was also found that samples of the same host species were sometimes grouped into the same clade. This suggests that there is some level of restructuring occurring at the hosts' species level. Beyond the *Polyrhachis* genus, there also appears to be separation of *Wolbachia* strains based upon being either Old World or New World, wherein taxa from the Old World will not typically be grouped into a closely related clade with New World taxa and vice versa.

Ultimately, the results of this study suggest that host biogeography and phylogeny do not have any significant impact on which strains of *Wolbachia* will be associated to the *Polyrhachis* host species, though our findings suggest that the *Polyrhachis* species level may have some effects on *Wolbachia* strain. Further work on the impact of geography of *Wolbachia* infection would benefit from incorporation of more data on the host's current range and historical biogeography, which were not included in this study. Additionally, horizontal transfer of *Wolbachia* between hosts is not common, but has been observed, primarily in related hosts [7]. Horizontal transfer events may affect the results of phylogenetic analyses, particularly in comparisons of the host's and *Wolbachia* phylogenies.

Our findings from this study, particularly our observation that some *Wolbachia* strains may be associated with particular *Polyrhachis* species, highlight the impacts that microbial diversity can have on ant diversity, and vice versa. The presence of vertically-transmitted symbionts like *Wolbachia* suggests the possibility of a microbial impact on evolution; coevolution of ants and microbes over long time-scales has already been observed in some ant genera, in some cases allowing the ants to pursue diets or occupy niches not previously available to them. While our findings about *Polyrhachis* help to elucidate more of the ways that symbionts can impact ant diversity, still, little is known about the microbial partners of the majority of ant genera. Studies of this nature are crucial in understanding the many factors that contribute to present-day ant diversity and may provide insights into the ways that the associates of ants may shape the evolutionary future of their hosts.

**Supplementary Materials:** The following supporting information can be downloaded at: https://www.mdpi.com/article/10.3390/d15030348/s1, File S1: Sample ID, host species, and country of origin for all for each of the initial 237 samples screened for *Wolbachia*. The presence of *Wolbachia* is indicated by the *wsp* column ("+" indicates positive for *Wolbachia*, "−" indicates negative for *Wolbachia*).

**Author Contributions:** Conceptualization, J.L.W., M.O.R., L.C.G. and C.S.M.; methodology, J.L.W., M.O.R. and L.C.G.; software, J.L.W., M.O.R. and L.C.G.; validation, J.L.W., M.O.R. and L.C.G.; formal analysis, J.L.W., M.O.R. and L.C.G.; investigation, J.L.W., M.O.R. and L.C.G.; resources, J.L.W., M.O.R. and L.C.G.; data curation, J.L.W. and M.O.R.; writing—original draft preparation, J.L.W.; writing—review and editing, L.C.G., M.O.R., C.S.M., and J.L.W.; visualization, J.L.W., M.O.R. and L.C.G.; supervision, M.O.R. and C.S.M.; project administration, C.S.M.; funding acquisition, C.S.M. and L.C.G. All authors have read and agreed to the published version of the manuscript.

**Funding:** This research was funded by National Science Foundation grant numbers DGE-1650441 and NSF DEB 1900357. The APC was waived by the journal.

**Institutional Review Board Statement:** Not applicable.

**Data Availability Statement:** New *Wolbachia* sequences are available on the National Library of Medicine's National Center for Biotechnology Information Sequence Read Archive at accession PRJNA937270.

**Acknowledgments:** JLW would like to thank Dirk Mezger and Corrie Moreau for the *Polyrhachis* DNA extractions used by this study. We would like to thank Benjamin Blanchard for this use of his new *Polyrhachis* phylogeny in our Mantel tests. We would also like to thank Corey Reese and Sam Cavanagh for their assistance and support in the early steps of this study. JLW also thanks the Cornell Institute of Host-Microbe Interactions and Disease (CIHMID) for giving her the opportunity to pursue this research as an intern and fellow. LCG thanks the National Science Foundation Graduate Research Fellowship for supporting part of this work (NSF DGE -1650441). CSM thanks the National Science Foundation for supporting part of this work (NSF DEB 1900357).

**Conflicts of Interest:** The authors declare no conflict of interest.

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
