# Peer review of "Investigating the Diversity of Wolbachia across the Spiny Ants (Polyrhachis)"

_diversity, doi:10.3390/d15030348_

Round 1

Reviewer 1 Report

Manuscript ‘Investigating the diversity of Wolbachia across the spiny ants (Polyrhachis)’ by Jenna L. Webb with co-authors presents a results of screening and multilocus characterization of Wolbachia symbionts in diversity of Polyrhachis hosts. The authors have managed to use in the analysis a significant part of species from Polyrhachis genus, sampling has been made over the World. The result can be briefly expressed as low Wolbachia diversity, Polyrhachis species mostly harbored closely related variants of Wolbachia. It is in accordance with some recent reports where limited diversity of Wolbachia found in a taxon (genus, family or even order). The big volume of data and scientific significance of the results allow to consider such material for publication in Diversity. However there are many lacks in the MS that  forces me to recommend ‘resubmission’.

Lines 30-32, 104-105 and further. The authors postulate a hypothesis that biogeography, phylogeny, and species level of Polyrhachis hosts are the key factors impacting on Wolbachia diversity. I can guess what authors meant however it should be explained for readers in detail. What results would confirm or decline the hypothesis. What scenarios could be?

Lines 54 – 56. ‘Since the initial identification of W. pipientis, most strains of the Wolbachia genus have been unable to be given species names due to a lack of understanding the genus’s genetic diversity.’ it is simplifying a complex issue.

Line 57. ‘…supergroups A to S’ I am afraid S is not the last.

Lines 98-100. ‘…Wolbachia infection in ants was illuminated and the biogeographical origin of the symbio- 99 sis was inferred to be in Asia’ with reference ‘7’. Do not cite this study in context of conclusion. Unfortunately this study was based on the wrong concept of Wolbachia symbionts were only in ants whereas infection in other arthropods was completely ignored.

Lines 117 – 118. ‘…and an elongation step at 70°C for 10 117 minutes.’ Why so long? Are you not afraid of false positive signals or unspecified bands?

Lines 170-171. How many species are infected in the collection?

Lines 187-188. Nested PCR can help.

lines 190-229. Too detail. Focus on unique alleles; for instance, synonymous or not (what class of aminoacid substitutions), p-distance between new and previously known allele. Present table of new alleles, namely Genbank, difference with the closest allele.

Table 1 is not necessary.

Table 2. Why hcpA and ftsZ datasets were combined while coxA and fbpA were subdivided?

Table 3. Sorting is in according on ID, it is wrong. It should be according to species or geography. Actually the data are analysed only for Wolbachia infected populations, so it seems the full table should transfer into SM.

Figure 1. I wonder to see 1) Sundaland, 2) Wallacean, and 3) Indomalayan on the same picture, first two are the subregions of the third region.

Figure 2. styling… what does mean ‘1996_WM206_F_Paratrechina_Mohali_Punjab’? or ’1869_Catr_A_A…’

Table 4. It is a great lack. Table 4 is misleading of readers, and line 256 can not resolve a problem. Let us see IDs MJ9280 and MJ9280 that according to the table are the same genotype but Figure 2 shows they are different. In addition table 4 and 5 should be combined.

Line 234 and further. The authors represents genotypes as distinct clades. How it could be??? Even a condition that authors use here, I mean ‘bootstrap values ≤ 70% were hidden’, indicates that they are not clades.

The discussion section taking into account the problems with conception, ‘clades’ etc may not consider.

Minor

Title: delete ‘investigating’

Authors: ‘…Corrie S. Moreau1 and 3’ ?

Abstract: delete ‘…not yet submitted to the MLST database.’

Introduction:

Lines 58-62. Considering G and R supergroups is not in focus of the study.

Line 72-73. underline?

Line 80. Open brackets.

Author Response

Reviewer 1

Manuscript ‘Investigating the diversity of Wolbachia across the spiny ants (Polyrhachis)’ by Jenna L. Webb with co-authors presents a results of screening and multilocus characterization of Wolbachia symbionts in diversity of Polyrhachis hosts. The authors have managed to use in the analysis a significant part of species from Polyrhachis genus, sampling has been made over the World. The result can be briefly expressed as low Wolbachia diversity, Polyrhachis species mostly harbored closely related variants of Wolbachia. It is in accordance with some recent reports where limited diversity of Wolbachia found in a taxon (genus, family or even order). The big volume of data and scientific significance of the results allow to consider such material for publication in Diversity. However there are many lacks in the MS that  forces me to recommend ‘resubmission’.

**Author response: Thank you for all comments! We have incorporated most of them in order to improve our manuscript.

Lines 30-32, 104-105 and further. The authors postulate a hypothesis that biogeography, phylogeny, and species level of Polyrhachis hosts are the key factors impacting on Wolbachia diversity. I can guess what authors meant however it should be explained for readers in detail. What results would confirm or decline the hypothesis. What scenarios could be?

**Author response: Thank you for this feedback; we agree that this point needs further clarification. We have added a paragraph at the end of the Introduction section explaining more about our hypotheses and what results would confirm or deny them.

Lines 54 – 56. ‘Since the initial identification of W. pipientis, most strains of the Wolbachia genus have been unable to be given species names due to a lack of understanding the genus’s genetic diversity.’ it is simplifying a complex issue.

**Author response: This section has been changed to add clarifying details about Wolbachia taxonomy and methods of assigning supergroup.

Line 57. ‘…supergroups A to S’ I am afraid S is not the last.

**Author response: This has been corrected and a new reference (reference 12) has been added.

Lines 98-100. ‘…Wolbachia infection in ants was illuminated and the biogeographical origin of the symbio- 99 sis was inferred to be in Asia’ with reference ‘7’. Do not cite this study in context of conclusion. Unfortunately this study was based on the wrong concept of Wolbachia symbionts were only in ants whereas infection in other arthropods was completely ignored.

**Author response:  The cited paper (Ramalho and Moreau 2020) talks about the origin of Wolbachia in ants–the authors don't focus on knowing the origin of Wolbachia in other insects. The authors Ramalho and Moreau 2020 did not assume that Wolbachia symbionts only occur in ants, it was just not the scope of that work to investigate the origin of Wolbachia in other insects. We changed in the main text it to make clear.

Lines 117 – 118. ‘…and an elongation step at 70°C for 10 117 minutes.’ Why so long? Are you not afraid of false positive signals or unspecified bands?

**Author response:  Thank you for noticing this. This may have been a typo: the actual elongation step was at 70°C for 10 minutes. This is consistent with the original Wolbachia MLST PCR protocol found in the Methods of Baldo et al. 2016, Applied and Environmental Microbiology.

Lines 170-171. How many species are infected in the collection?

**Author response: 69 of the 102 species were found to be infected. This information has also been added to the beginning of the results section.

Lines 187-188. Nested PCR can help.

**Author response: Thank you for this feedback. We will consider this new technique in further work.

lines 190-229. Too detail. Focus on unique alleles; for instance, synonymous or not (what class of aminoacid substitutions), p-distance between new and previously known allele. Present table of new alleles, namely Genbank, difference with the closest allele.

**Author response: We have added a haplotype network figure (Figure 3) which better illustrates the differences between alleles. We submitted sequences to GenBank and will have accession numbers available before this manuscript is published.

Table 1 is not necessary.

**Author response: The table has been removed and the information about annealing temperature was added to the methods section.

Table 2. Why hcpA and ftsZ datasets were combined while coxA and fbpA were subdivided?

**Author response: This partitioning scheme was determined via ModelFinder and is the same as was used in Ramalho and Moreau 2020, which was also published in Diversity.

Table 3. Sorting is in according on ID, it is wrong. It should be according to species or geography. Actually the data are analysed only for Wolbachia infected populations, so it seems the full table should transfer into SM.

**Author response: The full table with species negative for Wolbachia has been moved to Supplementary Materials 1. Table 2 (formerly Table 3) now only includes species that tested positive for Wolbachia, and is now organized alphabetically by species name.

Figure 1. I wonder to see 1) Sundaland, 2) Wallacean, and 3) Indomalayan on the same picture, first two are the subregions of the third region.

**Author response: Inclusion of the Sundaland and Wallacean subregions provides higher resolution in examining Polyrhachis’s geographic range. The “Indomalayan” group, in this case, more specifically describes the Indomalayan region north of Malaysia. This has been further explained in the figure legend. Additionally, Figure 1B shows a breakdown of region by country, so the reader will easily be able to determine the distinctions between each region.

Figure 2. styling… what does mean ‘1996_WM206_F_Paratrechina_Mohali_Punjab’? or ’1869_Catr_A_A…’

**Author response: The sample naming scheme has been changed in this latest version and can be translated as “sequence type_genus_species_country”. This has also been updated in the figure legend.

Table 4. It is a great lack. Table 4 is misleading of readers, and line 256 can not resolve a problem. Let us see IDs MJ9280 and MJ9280 that according to the table are the same genotype but Figure 2 shows they are different. In addition table 4 and 5 should be combined.

**Author response: Tables 4 and 5 have been combined into a new Table 3 and the figure legend has been changed. Additionally, the IDs referenced in the comment are the same ID (MJ9280). Assuming that Reviewer 1 mistyped MJ9280 and MJ9287, the table does not show them as the same genotype.

Line 234 and further. The authors represents genotypes as distinct clades. How it could be??? Even a condition that authors use here, I mean ‘bootstrap values ≤ 70% were hidden’, indicates that they are not clades.

The discussion section taking into account the problems with conception, ‘clades’ etc may not consider.

 **Author response: Thank you for this feedback. The 13 different genotypes we have assigned to the new Polyrhachis-associated Wolbachia strains (the context to which we believe the reviewer is referring) were once referred to as “clades” and now are exclusively referred to as “genotypes.” We use the term “clades” to discuss phylogenetic clades “a”, “b”, “b1”, “b2”, and “c”, featured on Figure 2.

Minor

Title: delete ‘investigating’

**Author response: Thank you for this suggestion. Though we have considered removing ‘investigating’ from the title, we have decided to leave it in as we feel it provides necessary description for the content of the study.

Authors: ‘…Corrie S. Moreau1 and 3’ ?

**Author response: Dr. Corrie S. Moreau is co-appointed in both the Entomology and Ecology and Evolutionary Biology departments at Cornell University.          

Reviewer 2 Report

The article " Investigating the diversity of Wolbachia across the spiny ants (Polyrhachis)" is devoted to the study of infection of species of this genus of ants with symbiotic Wolbachia bacteria, which are also widespread in other ant species. Correlations between the detection of symbionts and their diversity in the spiny ants with the phylogeny and geographical distribution of the host are considered.

The 102 sampled Polyrhachis species were taken to be representative of the entire host genus and spanned across the entire Polyrhachis biogeographical range. The authors then examined only some Wolbachia-positive specimens, although Wolbachia-positive host species not present in Mezger and Moreau's Polyrhachis phylogeny may also be of interest.

20 new strains of Wolbachia were found, in which the authors were able to identify all six genes of the conventional multilocus analysis. Another 14 strains have been identified by several bacterial genes. No correlation was found between both Wolbachia phylogenetic distance and geographic distance and Polyrhachis phylogenetic distance and Wolbachia phylogenetic distance. Although the authors found trends in the detection of related strains among representatives of the Polyrhachis genus. Another conclusion of this work is that spiny ant strains of Wolbachia can cluster depending on whether the hosts are from the Old World or the New World.

The such study design is particularly interesting and could bring important findings about co-evolutionary relations between symbiotic bacteria and ants. I have several comments, which should be considered or explained by the Authors in the revision of their work:

Major

All nucleotide differences should be shown in all new Wolbachia allele variants. It is clear from the text that they are not identical in each of the alleles. All of them must be annotated at least in GenBank. Differences in the nucleotide composition of each new allele from the already known should be shown in the article for all 34 identified strains. Perhaps in an additional table or in haplotype nets for each gene.

Table 3 should be moved to the Supplementary Material of the article. Instead, I would recommend compiling another table, for example: species (columns on the left), their occurrence in countries (columns above) and Wolbachia infection (in brackets in the columns of the table).

An example of Wolbachia from supergroup B strain from ant host Pheidole sciophila should be added to Fig 2.

Clade b2 consist only New World samples but colored as Old World. This does not match the Figure caption and is confusing.

Minor

remove the link to the MLST database from the abstract

L95 missing dots at the end of a sentence

L210  produced and inde-  error

Author Response

Reviewer 2

The article " Investigating the diversity of Wolbachia across the spiny ants (Polyrhachis)" is devoted to the study of infection of species of this genus of ants with symbiotic Wolbachia bacteria, which are also widespread in other ant species. Correlations between the detection of symbionts and their diversity in the spiny ants with the phylogeny and geographical distribution of the host are considered.

The 102 sampled Polyrhachis species were taken to be representative of the entire host genus and spanned across the entire Polyrhachis biogeographical range. The authors then examined only some Wolbachia-positive specimens, although Wolbachia-positive host species not present in Mezger and Moreau's Polyrhachis phylogeny may also be of interest.

20 new strains of Wolbachia were found, in which the authors were able to identify all six genes of the conventional multilocus analysis. Another 14 strains have been identified by several bacterial genes. No correlation was found between both Wolbachia phylogenetic distance and geographic distance and Polyrhachis phylogenetic distance and Wolbachia phylogenetic distance. Although the authors found trends in the detection of related strains among representatives of the Polyrhachis genus. Another conclusion of this work is that spiny ant strains of Wolbachia can cluster depending on whether the hosts are from the Old World or the New World.

The such study design is particularly interesting and could bring important findings about co-evolutionary relations between symbiotic bacteria and ants. I have several comments, which should be considered or explained by the Authors in the revision of their work:

**Author response: Thank you for the feedback! We have incorporated most of your comments to improve the manuscript.

Major

All nucleotide differences should be shown in all new Wolbachia allele variants. It is clear from the text that they are not identical in each of the alleles. All of them must be annotated at least in GenBank. Differences in the nucleotide composition of each new allele from the already known should be shown in the article for all 34 identified strains. Perhaps in an additional table or in haplotype nets for each gene.

**Author response: We have added in a haplotype network figure (Figure 4) that better illustrates differences between the new Wolbachia allele variants. Further, we have begun the process of submitting all sequences to GenBank. We expect to include accession numbers in this manuscript before publication.

Table 3 should be moved to the Supplementary Material of the article. Instead, I would recommend compiling another table, for example: species (columns on the left), their occurrence in countries (columns above) and Wolbachia infection (in brackets in the columns of the table).

**Author response: Thank you for this suggestion. One of the other reviewers suggested including only the taxa with positive Wolbachia infections in this table and moving the larger table to the Supplementary Material. We have done so at their suggestion and to more easily communicate the number of positively infected samples (as opposed to showing which species and countries had the most Wolbachia infections). This is Table 2 in the newest version.

An example of Wolbachia from supergroup B strain from ant host Pheidole sciophila should be added to Fig 2.

**Author response: The taxon was included previously but excluded because Wolbachia sequences from Pheidole sciophila were the only ones that represented supergroup B. More sequences from other ant taxa are necessary to determine that this is indeed a valid clade and has not be misidentified due to error or sampling bias. Ramalho and Moreau 2020 (also published in Diversity) also excluded this taxon for the same reasons.

Clade b2 consist only New World samples but colored as Old World. This does not match the Figure caption and is confusing.

**Author response: Thank you for this suggestion. The b2 clade has been changed to a color distinct from the other clades.

Minor

remove the link to the MLST database from the abstract

**Author response: The link has been removed.

L95 missing dots at the end of a sentence

**Author response: The punctuation has been added.

L210  produced and inde-  error

**Author response: In this case, “an” was mistyped as “and.” This has now been corrected.

Reviewer 3 Report

The authors ask about the genetic diversity and potential co-evolution between Wolbachia and its hosts belonging to the spiny ants (genus Polyrhachis). The research question is current and interesting, especially taking into account the wide spatial scale of the described study. The analyses of the prevalence of infections with Wolbachia among invertebrates, as well as the further studies focused on Wolbachia's impact on host biology are of broad interest.

I recommend the submitted manuscript for publication, but I have a few remarks that I believe will help to improve the overall presentation of the manuscript.

Q1 – Please describe more precisely the protocol of DNA extraction (i.e. if whole ants' bodies were homogenized prior to the extraction or the DNA was extracted only from abdominal tissues).

Q2 – Have you wondered why the amplification of some MLST fragments failed during the study? (e.g. mutations in primers binding sites or low level of DNA template).

Q3 – in Table 4 authors indicated new alleles and their close matching STs according to the MLST database. Have you added those alleles to the MLST database or any other open database (e.g. GenBank)? I did not find any details on accession numbers and data availability.

Q4 – many samples had to be rejected due to the multiple infections with different Wolbachia strains. Do you plan to continue your study and resolve this limitation?

Q5 – tables are massive and thus less readable (especially Table 3). Could you please rearrange them or turn them into supplementary data?

Q6 – figures should be placed close to the text where they are mentioned. Please correct.

Additional comments:

1.      line 48 – a space should be added before the reference [5]

2.      lines 72-73 – there is an underline added; please correct

3.      line 81 – a space should be added before references [23,24]

4.      line 86 – a space should be added before references [27,28]

5.      line 95 – the lack of a dot at the end of the sentence

6.      line 105 – additional spaces, please correct

7.      line 141 – a space should be added before the reference [34]

Author Response

Reviewer 3

the authors ask about the genetic diversity and potential co-evolution between Wolbachia and its hosts belonging to the spiny ants (genus Polyrhachis). The research question is current and interesting, especially taking into account the wide spatial scale of the described study. The analyses of the prevalence of infections with Wolbachia among invertebrates, as well as the further studies focused on Wolbachia's impact on host biology are of broad interest.

I recommend the submitted manuscript for publication, but I have a few remarks that I believe will help to improve the overall presentation of the manuscript.

**Author response: Thank you for your feedback and interest in our manuscript! We have incorporated the majority of your suggested edits.

Q1 – Please describe more precisely the protocol of DNA extraction (i.e. if whole ants' bodies were homogenized prior to the extraction or the DNA was extracted only from abdominal tissues).

**Author response: Whole ant bodies were homogenized before extracting DNA. This detail has been added to the methods section.

Q2 – Have you wondered why the amplification of some MLST fragments failed during the study? (e.g. mutations in primers binding sites or low level of DNA template).

**Author response: We’ve definitely wondered about this! There is often differential amplification between different samples and different genes. Sometimes lower amplification is most likely due to poor quality of DNA template, but other times it is less easy to explain. Hopefully, further study will give us more insight as to what causes amplification failure for different fragments.

Q3 – in Table 4 authors indicated new alleles and their close matching STs according to the MLST database. Have you added those alleles to the MLST database or any other open database (e.g. GenBank)? I did not find any details on accession numbers and data availability.

**Author response: Thank you for pointing this out. We have begun the process of adding the new sequences to GenBank and we expect to have accession numbers available for all sequences before this study is published.

Q4 – many samples had to be rejected due to the multiple infections with different Wolbachia strains. Do you plan to continue your study and resolve this limitation?

**Author response: The first author on this paper, Jenna Webb, was an undergraduate while working on the paper materials. She has since begun a graduate program studying a different topic. This group of authors will likely not continue this particular study, though we certainly welcome others in the field to improve upon this work.

Q5 – tables are massive and thus less readable (especially Table 3). Could you please rearrange them or turn them into supplementary data?

**Author response: The former Table 3 has been moved to the Supplementary Materials, and a new table (now Table 2) with only samples positive for Wolbachia has been added in its place.

Q6 – figures should be placed close to the text where they are mentioned. Please correct.

Additional comments:

  1. line 48 – a space should be added before the reference [5]

**Author response: Thank you. This has been corrected.

  1. lines 72-73 – there is an underline added; please correct

**Author response: We are unable to find the underline specified in lines 72-73.

  1. line 81 – a space should be added before references [23,24]

**Author response: The space was added.

  1. line 86 – a space should be added before references [27,28]
  2. line 95 – the lack of a dot at the end of the sentence
  3. line 105 – additional spaces, please correct
  4. line 141 – a space should be added before the reference [34]

**Author response: Comments 4-7 were corrected. Thank you for the feedback.

Round 2

Reviewer 1 Report

The second variant of MS is noticeably bettet than previous one however it still has serious lacks.

I am sure that analysis should be improved, text could be drastically condensed and clarity to enhance.

1. The authors postulated at the begining three hypothesis: genetic pattern affected by host phyloigeny, geography or species level. In present version the authors inserted a part (104-116 lines) that clarify it, that is very good. However I can not understand „If species level within Polyrhachis impacts the observed diversity of Wolbachia, we may observe different kinds of Wolbachia infecting different Polyrhachis species in statistically significant ways.“ As to geography hypothesis the reader needs information about hystory of these species. How they expanded on this territory, how long it lived here, what range of each species. It is necessary to realise how a local sample of a certain species give a piece to a common inference of ant-Wolbachia association. And finally about phylogeny. The authors found many double or multi-infected cases. What nature of these phenomenon? One possibility is Wolbachia divergence within ant lineage. Such facts are known but they are not numerous. This is ok for phylogeny hypothesis used here, but it should be demonstrated (low divergence). Another way of explanation is horisontal transmission of Wolbachia. The last possibility (numerous facts) cancel phylogeny hypothesis at once. Ok, lets consider fantasy case that there are no double infected associations. To test a phylogeny hypothesis the authors should compare phylogeny tree of Wolbachia with Polyarhis ones. So, I think the authors could consider geography and phylogeny ideas (species level I dont understand) in discussion but it should not postulated as the main line.

2. ST.

The authors combined the tables of MLST, it is ok. However I again see many identical isolates whereas in the MS body indicated two tens new ST. I see coloured table, nonsence. You can use colour in figure, but in table a text and a structure are the main. Please read papers of other researchers and found out how they solve a problem with MLST data in a period when new data could not be deposited.

3. Wolbachia phylogeny

Look at figure 2. Clades «a», «b» and «c» have statistical support less than 70. It means >30% of ML-trees have no such clades. In adittion, look how small distances between anounced clades (inner brances). I can not seriously consider idea of these clades.

4. The analysis.

Partially it crosses with doi:10.3390/d12110426 that was mentioned in first review and in the author response. We know that Wolbachia is not only transfer between hosts but even recombinate, making traces of distant related Wolbachia lineages. The authors should consider not only bacterial diversity of ants but any other variants that have the same or close-relative allels.

5. There are huge amount repetitions in the MS. Many portions could be expressed much shorter. Many paragraphs can be restructered. For instance, 199-237 lines are narrative and have many repetitiones. It may be restructured like «We found 100 uniqe alleles, 70 of them are new. The number of alleles is grow in order coxA-gatB-ftsZ-hcpA-fbpA. However genetic distances between allels were highest in ... Most of SNP does not lead to amino-acid replacements...». Such information as «Of the 15 samples appearing to have new ftsZ allele variants, only six (P. cephalotes [Malaysia], P. Myrma sp. [Uganda], P. thrinax [India], P. [Myrma]sp. [Tanzania], Polyrhachis sp. [Singapore], P. hexacantha [Australia])» should be deleted. There is in the Table 2, and in the text the authors can point only important information. Disscussion section is too big and boring to read up.

Minor comments on Introduction section

- 34-35 lines, «all 20 are seemingly new strains not yet submitted to the MLST database.» delete. And why seemingly? Did you check blastN?

- 46 line, «male-killing or feminization,» why or?

- 47 line, open brackets

- 54 line, «as a single species». What about doi:10.3390/ijms21218064? I recomend do not touch topic of Wolbachia species at all.

- Lines 57-60 too detail, it is not in focus

- 63-66 lines. Probably something is known about Wolbachia patterns beyond the ants?

- 80-81 and 90 lines, repetition.

Next time I wish to read this study in a Journal not in manuscript format. Again, material is great, text is poor, analysis has lacks.

Author Response

The second variant of MS is noticeably bettet than previous one however it still has serious lacks.

I am sure that analysis should be improved, text could be drastically condensed and clarity to enhance.

**Author response: Thank you for this additional feedback. We have incorporated many of your suggestions into the latest version of the manuscript.

  1. The authors postulated at the begining three hypothesis: genetic pattern affected by host phyloigeny, geography or species level. In present version the authors inserted a part (104-116 lines) that clarify it, that is very good. However I can not understand „If species level within Polyrhachis impacts the observed diversity of Wolbachia, we may observe different kinds of Wolbachia infecting different Polyrhachis species in statistically significant ways.“ As to geography hypothesis the reader needs information about hystory of these species. How they expanded on this territory, how long it lived here, what range of each species. It is necessary to realise how a local sample of a certain species give a piece to a common inference of ant-Wolbachia association. And finally about phylogeny. The authors found many double or multi-infected cases. What nature of these phenomenon? One possibility is Wolbachia divergence within ant lineage. Such facts are known but they are not numerous. This is ok for phylogeny hypothesis used here, but it should be demonstrated (low divergence). Another way of explanation is horisontal transmission of Wolbachia. The last possibility (numerous facts) cancel phylogeny hypothesis at once. Ok, lets consider fantasy case that there are no double infected associations. To test a phylogeny hypothesis the authors should compare phylogeny tree of Wolbachia with Polyarhis ones. So, I think the authors could consider geography and phylogeny ideas (species level I dont understand) in discussion but it should not postulated as the main line.

**Author response: Thank you again for your earlier suggestion to clarify our hypotheses on Wolbachia’s impact on the Polyrhachis phylogeny, geography, and species level. We feel that these clarifications strengthen the manuscript considerably.

This comment has several components which we address here one by one:

  • The referenced passage “If species level within Polyrhachis…” means that we were testing to see if the same Polyrhachis species is more likely to have similar Wolbachia infections than different Polrhachis species.
  • We agree that analyses of Wolbachia and host geography would benefit from more information about the host species history. We have now addressed this limitation in the discussion section.
  • We have addressed horizontal transmission of Wolbachia in reviewer comment 4 and have added clarification in the discussion section.
  • We agree that it is necessary to test the phylogeny hypothesis with a comparison of the host and Wolbachia phylogenetic trees. In fact, the second Mantel test we used tested correlations between the Polyrhachis and Wolbachia phylogenies. Because the results did not indicate a correlation, we found it unnecessary to do any other phylogenetic tests.

The authors combined the tables of MLST, it is ok. However I again see many identical isolates whereas in the MS body indicated two tens new ST. I see coloured table, nonsence. You can use colour in figure, but in table a text and a structure are the main. Please read papers of other researchers and found out how they solve a problem with MLST data in a period when new data could not be deposited.

**Author response: To clarify, the blue shaded cells indicate new allele variants for each loci. Many of the samples share the same closest-matching ST but still have differences in allele variants.

  1. Wolbachia phylogeny

Look at figure 2. Clades «a», «b» and «c» have statistical support less than 70. It means >30% of ML-trees have no such clades. In adittion, look how small distances between anounced clades (inner brances). I can not seriously consider idea of these clades.

**Author response: We understand the reviewer's frustration. In fact, we would love for our analyses to returned more robust clades. However, although they do not pass the 70% cut-off, these samples are still grouped with some robustness. For the current study, our main finding places all samples of Polyrchachis belonging to the same Wolbachia supergroup A, and almost always associated with other Wolbachia found in Old Word. In addition, previous papers (Russell et al 2009, 2012, Ramalho & Moreau 2020) also found evidence of distinct Wolbachia between the American continent and the Old World. So, despite the bootstrap values not being above 70%, our results were still consistent.

  1. The analysis.

Partially it crosses with doi:10.3390/d12110426 that was mentioned in first review and in the author response. We know that Wolbachia is not only transfer between hosts but even recombinate, making traces of distant related Wolbachia lineages. The authors should consider not only bacterial diversity of ants but any other variants that have the same or close-relative allels.

**Author response: We appreciate this comment and understand that host transfer of Wolbachia between ants and other insects is possible (though seemingly more common in related hosts).

  1. There are huge amount repetitions in the MS. Many portions could be expressed much shorter. Many paragraphs can be restructered. For instance, 199-237 lines are narrative and have many repetitiones. It may be restructured like «We found 100 uniqe alleles, 70 of them are new. The number of alleles is grow in order coxA-gatB-ftsZ-hcpA-fbpA. However genetic distances between allels were highest in ... Most of SNP does not lead to amino-acid replacements...». Such information as «Of the 15 samples appearing to have new ftsZ allele variants, only six (P. cephalotes [Malaysia], P. Myrma sp. [Uganda], P. thrinax [India], P. [Myrma]sp. [Tanzania], Polyrhachis sp. [Singapore], P. hexacantha [Australia])» should be deleted. There is in the Table 2, and in the text the authors can point only important information. Disscussion section is too big and boring to read up.

**Author response: We agree that the referenced sections in the results were unnecessarily long and have been condensed for clarity and readability. The discussion has been left as-is; we feel that the interpretation of results within the discussion is necessary for the reader’s understanding.

Minor comments on Introduction section

- 34-35 lines, «all 20 are seemingly new strains not yet submitted to the MLST database.» delete. And why seemingly? Did you check blastN?

**Author response: The section “..not yet submitted to the MLST database” was deleted. These appear to be new strains after checking against reference sequences in the MLST database (see Methods, lines 134-135).

- 46 line, «male-killing or feminization,» why or?

**Author response: Thank you for pointing this out. This has been corrected to “…male killing, male feminization, and…”

- 47 line, open brackets

**Author response: Thank you, this error was corrected.

- 54 line, «as a single species». What about doi:10.3390/ijms21218064? I recomend do not touch topic of Wolbachia species at all.

**Author response: The mention of Wolbachia species was removed from the manuscript.

- Lines 57-60 too detail, it is not in focus

**Author response: Thank you for this feedback and we agree—the sentence referencing Wolbachia outside of ants was removed.

- 63-66 lines. Probably something is known about Wolbachia patterns beyond the ants?

**Author response: We are confused by this comment in context of the lines referenced. Line 63 is at the end of a paragraph discussing Wolbachia supergroups in ants—previously context about other insects was included, but taken out (in response to the comment right above this one on lines 57-60). Lines 64-66 are about Wolbachia sequence typing with wsp and does not mention ants. But, in response to this question, Wolbachia patterns are known beyond the ants, but it is outside the scope of this paper to describe these patterns in depth.

- 80-81 and 90 lines, repetition.

**Author response: We appreciate this feedback, but we do not feel this is repetitive as the sections referenced bring up the ant tribe Camponotini and genus Cephalotes, distinct groups with variable relationships with Wolbachia and other symbiotic microbes.

Next time I wish to read this study in a Journal not in manuscript format. Again, material is great, text is poor, analysis has lacks.